# Mapping of Olive Trees Using Pansharpened QuickBird Images: An Evaluation of Pixel- and Object-Based Analyses

**Isabel Luisa Castillejo-González** 

Department of Graphic Engineering and Geomatics, University of Cordoba, Campus de Rabanales, 14071 Córdoba, Spain; ilcasti@uco.es; Tel.: +34-957-218-537

**Abstract:** This study sought to verify whether remote sensing offers the ability to efficiently delineate olive tree canopies using QuickBird (QB) satellite imagery. This paper compares four classification algorithms performed in pixel- and object-based analyses. To increase the spectral and spatial resolution of the standard QB image, three different pansharpened images were obtained based on variations in the weight of the red and near infrared bands. The results showed slight differences between classifiers. Maximum Likelihood algorithm yielded the highest results in pixel-based classifications with an average overall accuracy (OA) of 94.2%. In object-based analyses, Maximum Likelihood and Decision Tree classifiers offered the highest precisions with average OA of 95.3% and 96.6%, respectively. Between pixel- and object-based analyses no clear difference was observed, showing an increase of average OA values of approximately 1% for all classifiers except Decision Tree, which improved up to 4.5%. The alteration of the weight of different bands in the pansharpen process exhibited satisfactory results with a general performance improvement of up to 9% and 11% in pixel- and object-based analyses, respectively. Thus, object-based analyses with the DT algorithm and the pansharpened imagery with the near-infrared band altered would be highly recommended to obtain accurate maps for site-specific management.

**Keywords:** Á Trous algorithm; conservation agriculture; crop inventory; remote sensing; spectral-weight variations in fused images

## 1. Introduction

Nowadays, one of the most important objectives in agriculture is to perform precision agriculture (PA) in most possible scenarios to control efficiently the input data and, consequently, reduce the production cost and the environmental pollution produced by this activity. To facilitate and assist this change in farm management, government institutions tend to regulate and encourage different techniques based on PA. The European Commission is greatly concerned about the new challenges in agriculture and promote the changes by different legal instruments and key texts [1]. Diverse action areas such as farming, protection of natural or agricultural environment, food safety, security and traceability, or even climate change mitigation are regulated. Some of the recommended or mandatory practices are supported in an accurate control of the spatial distribution of crops. To obtain economical funds from the common agricultural policy [2], PA promotes among many other actions, the application of site-specific management or integrated management systems in crops production to reduce the use of fertilisers, herbicides, or pesticides and the establishment of certain conservation agro-environmental measures such as cover crops in olive orchards [3]. To control the correct application of PA techniques, different monitoring systems were developed. The expensive, time-consuming, and imprecise system based on sample and ground visit to small percentage of fields has forced a search for new

techniques that reduce costs and increase the controlled area, maintaining high accuracy in the analyses. Remote sensing data can significantly improve the deficiencies of ground visits, allowing accurate maps.

Several studies have focused on addressing diverse PA topics by remote sensing, such as obtaining accurate maps of crops [4–6]; detecting the location of weeds [7–9], pests [10–12], and diseases [13–15] to apply site-specific management, or determining the level of water stress to design optimal irrigation systems [16–18]. Nevertheless, most of the precision agriculture studies with remote sensing analysed the characteristics of herbaceous crops, as these types of crops usually cover all the field and are easier to study with digital imagery. Woody crops present very different spectral responses between tree canopies, soils, and other covers presented in the field. Thus, very high spatial resolution images are needed. Most of the studies which aim is to characterize the architecture of the trees used airborne or Unmanned Aerial Vehicle (UAV) images to obtain the canopy information [19–21]. Nevertheless, few studies used satellite imagery [22–24] and most of them were aid with LiDAR information [25–27].

High resolution satellite imagery can be useful to accurately map tree canopies. Companies that distribute Earth Observation Satellite images usually offer the user community two separate products: a high-spatial resolution panchromatic image and a low-spatial resolution multispectral one. While the multispectral image facilitates the discrimination of land covers types, the panchromatic image allows to delimit accurately each land cover [28]. To use simultaneously the advantages of both resolutions in one image, fusion techniques have been developed. The pansharpen fusion method allows the injection of spatial detail information from the panchromatic image into each band of the multispectral image [29]. These new characteristics of the pansharpened image can help to accurately delineate the tree canopies to apply PA techniques.

Supervised classification methods are extensively used in land use classification studies [30]. These procedures extrapolate the spectral characteristics obtained from the image training sites defined for classification by the user to other areas of the image. Nowadays, there are classification routines based on spectral or angular distances, probability analysis, and more advanced data mining techniques. There is no one ideal classification routine. The most appropriate method is determined by the needs and requirements of each study [31]. Many of the remote sensing classifications are based on pixels as the minimum spatial information unit. These analyses provide very good results in homogeneous land uses. Nevertheless, the increase of spatial resolution causes an increase in the intraclass spectral variability and a reduction in classification performance and accuracy when pixel-based analyses are used [32]. This is particularly true when the pixel size is significantly smaller than the average size of the objects of interest [33]. To overcome this problem, different segmentation techniques, in which adjacent pixels are grouped into spectrally and spatially homogeneous objects, have been developed. The main segmentation algorithms can be classified into two general classes: edge-based and object-merging algorithms [34]. Most of the segmentation procedure developed are object merging algorithms, which take some pixels as seeds and grow the regions around them based on certain homogeneity criteria [35]. Since Kettig and Landgrebe [36], the object-based approach has hardly been used in favour of easier pixel-based analyses. Some researchers have reported that the segmentation techniques used in classifications reduce the local variation caused by textures, shadows and shape in forestry trees [37,38] and agricultural trees [39] classifications. However, object-based classifications in typical agricultural dryland Mediterranean areas to map accurately olive tree canopies using only high spatial resolution satellite imagery are lacking.

Therefore, the main objective of this paper was to evaluate the potential of four supervised classification routines, applied to pixel- and object-based classifications, to delineate olive tree canopies using a pansharpened QuickBird image. An additional goal was to check the effect of the variation of the spectral weight in the pansharpen process, to emphasise the spectral information of different wavelengths over another.

## 2. Materials and Methods

### 2.1. Study Area and Satellite Image Acquisition

This study was focused on dryland olive (*Olea europaea* L.) orchards representative of the typical continental Mediterranean climate [40]. The analysis was conducted in five olive orchards fields named A, B, C, D and E (Figure 1) located near Montilla, province of Córdoba (Andalusia, southern Spain, centre UTM *X-Y* coordinates 355,746–4,164,520 m, datum WGS84). This agricultural region has a typical continental Mediterranean climate characterized with short mild winters and long dry summers [41]. The study sites were located in a farmer-managed area where the farmers made decisions individually. Thus, different characteristics of the studied fields such as size and morphology of olive crowns, presence or not of vegetation cover or soil tillage, were found. The total areas of the A, B, C, D and E fields were 2.16 ha, 22.80 ha, 15.66 ha, 24.55 ha and 28.44 ha, respectively.

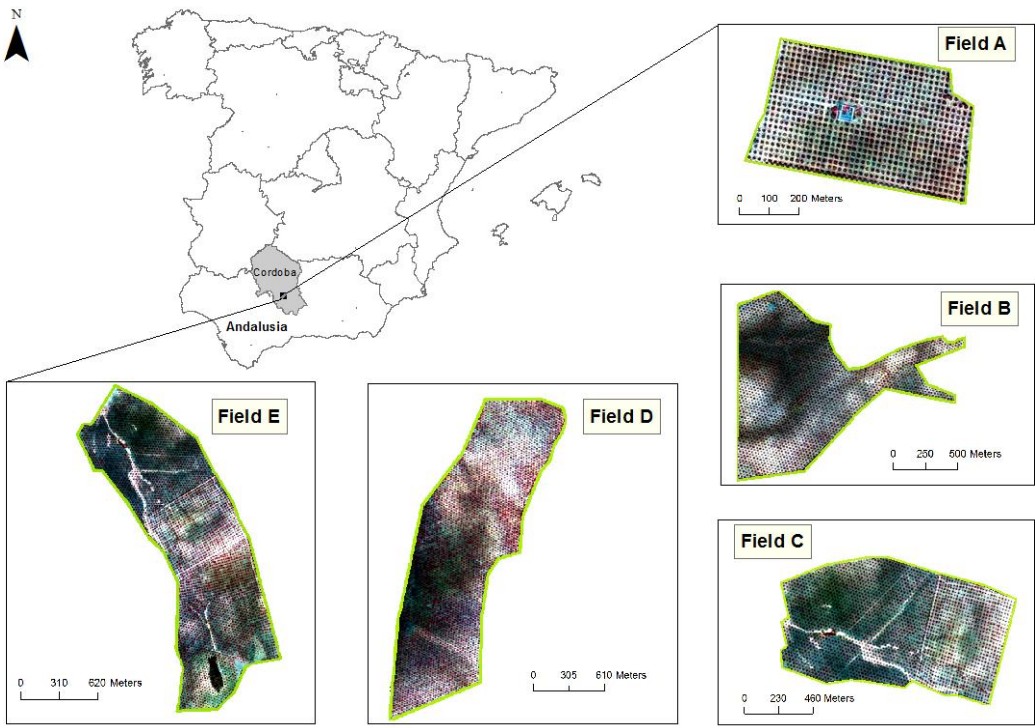

**Figure 1.** Location of the study area in Andalusia, southern Spain. Detailed olive orchards fields are depicted by QuickBird pansharpened images.

On 10 July 2004, a QuickBird (QB) satellite digital image was acquired for the study area. The QB satellite provided four multispectral bands (blue, B: 450–520 nm; green, G: 520–600 nm; red, R: 630–690 nm; and near-infrared, NIR: 760–900 nm) with a spatial resolution of 2.8 m, and a panchromatic band (PAN: 450–900 nm) with a spatial resolution of 0.7 m. Radiometric resolution of the QB image was 11 bit. A QB Standard image product was ordered, which included radiometric, sensor and geometric corrections previously carried out by the distributor [42]. No atmospheric corrections were needed as long as each classification was carried out in a single date image on the same relative scale [43].

### 2.2. Data Fusion: Pansharpening of Multispectral Images

To obtain an image of high spectral and spatial resolution, a pansharpening process was carried out with the QB bands. The pansharpen techniques allow to obtain new bands with the spectral resolution of the multispectral bands and the spatial resolution of the panchromatic band. In this study, a weighting variant of the fusion algorithm based on the wavelet transform calculated using the Á Trous algorithm was used to fuse the multispectral and panchromatic bands [44]. This fusion

method consists basically of successive convolutions between the analyzed image and a low-pass filter called the scaling function, which commonly is the b3-spline function. The filters to be applied in the subsequent decomposition levels are obtained from the filter applied in the previous level, intercalating it with zeros in the rows and columns. The wavelet coefficients are obtained from the difference between two consecutive decomposition levels.

Knowing that the red and especially the near infrared bands are very important in the discrimination of vegetation, these two bands were weighted differently. Thus, to evaluate the effect of different weighted pansharpened bands in the classifications, three weight combinations were proposed: (a) 1-1-1-1, (b) 1-1-5-5 and (c) 1-1-1-10 as weight factor for B-G-R-NIR bands, respectively. As a result of this fusion or pansharpening process, three different images with four multispectral bands (B, G, R and NIR) and with a spatial resolution of 0.7 m were obtained.

Usually, the global quality of the resulting pansharpened image is estimated with the ERGAS index (Erreur Relative Globale Adimensionnelle de Synthèse) [45]. This relative dimensionless global error in synthesis offers a global picture of the spectral quality of the fused product. Nevertheless, in pansharpening techniques, high spectral quality implies low spatial quality and vice versa, which suggests the necessity to control not only the spectral quality of the process, but also the spatial. Thus, a spatial index based on ERGAS concepts and translated to the spatial domain [46] was used.

### 2.3. Segmentation

A segmentation procedure was performed to partition the QB pansharpened images into homogeneous objects using the Fractal Net Evolution Approach (FNEA) segmentation algorithm [47] (Figure 2). This algorithm allows the multiresolution bottom-up region-merging segmentation, a process in which individual pixels merge to objects in successive fusing iterations. The merging process continues until a threshold derived from the user-defined parameters is reached. The result is an image in which the pixels are aggregated in highly homogeneous objects at an arbitrary resolution.

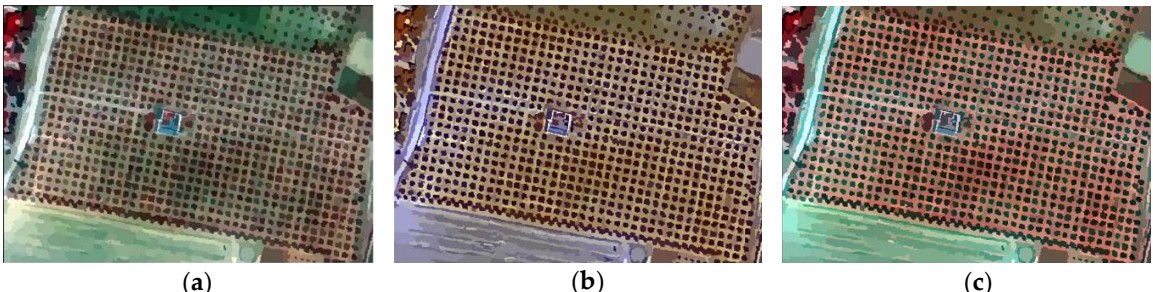

|        |        |        |
|:------:|:------:|:------:|
| (**a**) | (**b**) | (**c**) |

**Figure 2.** Multiresolution segmentation of pansharpened QuickBird (QB) imagery in field A. Pansharpen weight (B-G-R-NIR): (**a**) 1-1-1-1; (**b**) 1-1-5-5; and (**c**) 1-1-1-10.

The segmentation process can be controlled by the weighting of the input data and the definition of three parameters. The scale parameter controls the size of the objects, while the colour and shape parameters define the importance of the spectral and morphological information involved in the object generation, respectively. The setting of the segmentation parameters were determined by testing different segmentation input scenarios to evaluate their ability to delineate olive crowns. For each field, the first parameter to adjust was the scale parameter to control the size of the objects depending on the characteristics of the field. Then, with the scale parameter fixed, the spectral and morphological weight of the information was defined. The morphological information is divided into two characteristics, the compactness and the smoothness of the objects. For this study, as the trees crowns present a compact structure, these two characteristics were fixed in all scenarios tested to 0.8 and 0.2 for compactness and smoothness, respectively.

The segmentation procedure generates not only the mean spectral information of the objects, which is derived from the spectral information of the pixels that form each object, but also a large

amount of data divided mainly in three categories: spectral, morphological and textural. In this study, only some spectral and morphological variables derived from the segmentation process were used to characterize the olive orchards. Therefore, to perform this analysis eight object-based features, three spectral and five morphological variables, were included (Table 1). The spectral feature *Mean* is calculated for each multispectral band independently, obtaining 4 final features: Mean (Blue), Mean (Green), Mean (Red) and Mean (Near-Infrared).

**Table 1.** Object-based features derived from segmentation.

| Categories | Features | Brief Description |
| --- | --- | --- |
| **Spectral** | Mean | Mean of the intensity values of all pixels forming an image object |
| | NDVI | Normalized Difference Vegetation Index [48] |
| | RDVI | Renormalized Difference Vegetation Index [49] |
| **Shape** | Area | Number of pixels forming an image object |
| | Asymmetry | Relative length of an image object compared to a regular ellipse polygon |
| | Border index | Ratio between the border lengths of the image object and the smallest enclosing rectangle |
| | Length | Multiplication between the number of pixels and the length-to-width ratio of an image object |
| | Width | Ratio between the number of pixels and the length-to-width ratio of an image object |

## 2.4. Classification and Accuracy Assessments

Both pixel- and object-based analyses with four different supervised classifiers were conducted on the five olive orchards fields. The classification algorithms were Minimum Distance (MD), Spectral Angle Mapper (SAM), Maximum Likelihood (ML) and Decision Tree (DT). MD classifies each pixel in the category that presents the minimum spectral distance between the spectral signal of the pixel and the spectral average of the class. The spectral distance is determined by Euclidean distance in N-bands spectral space [50]. Similarly, SAM measures spectral similarity but assigns the category of each pixel to the class that presents the minimum spectral angle, instead the spectral distance. The spectral angle between two spectra is calculated by taking the arccosine of the dot product of the two spectral vector [51]. ML creates classification rules based on probabilistic algorithms considering the spectral average of each class and the variance. This algorithm assigns a pixel to the most probable class and thus minimizes the probability of error using Bayesian theory [52]. Finally, DT classifier creates models of decision based on conditional control statements. In this study, the DT classification was performed with the data mining C4.5 algorithm, a top-down inductor of decision trees that expands nodes in depth-first order for each step using the divide-and-conquer strategy [53]. Ground-truth land use was randomly defined to substantiate and validate the classification procedures. For each field, a sampling with distant and independent locations was digitized directly on the image. Approximately 25% of the sampled surface were used to collect the spectral signature in the training process, and the remaining 75% were used to assess the accuracy of the classifications. To avoid any subjective estimation, the training and verification procedure did not change in any of the classifications.

To determine the accuracy obtained with every classifier in each olive orchard field, the confusion matrix of the classification and the Kappa test were analysed. The confusion matrix compares the percentage of classified pixels of each class with the verified ground truth class, indicating the correct assessment and the errors among the studied classes [38]. In addition to detailed accuracies obtained in every classification category, the confusion matrix obtain the overall accuracy (OA), which indicates the overall percentage of correctly classified pixels in the classification. The Kappa test yields the Kappa coefficient (K), which determines if the results obtained in the classification are significantly better than the results obtained in a random classification. The combination of both accuracy values is more conservative than a simple percent agreement value [54,55].

In pixel-based classification it is frequent to observe isolated-misclassified pixels dispersed within the area of another class. To reduce this commonly named salt and pepper noise and increase the accuracy of the classifications, a majority filter of 3 × 3 was applied to improve the classified maps. In object-based classification this noise is practically eliminated when pixels group in the segmentation process.

The pansharpened QB bands were generated with the IJFUSION software (Polytechnic University of Madrid, Spain). To obtain the segmented bands used in object-based classifications, the eCognition Developer 8 software (Definiens AG) was used. The Weka 3.8 software (University of Waikato, New Zealand) determined the decision tree sequences. Finally, the software ENVI 5.1 (Harris Geospatial Solutions) was used to carry out all the pixel- and object-based classifications.

## 3. Results and Discussion

### 3.1. Data Fusion: Pansharpening of Multispectral Images

Three pansharpen procedures were performed with the entire QB image. To control the quality of the process, two ERGAS indexes were calculated. Table 2 shows the spectral and spatial ERGAS indexes obtained in the study for each pansharpen combination. The spectral ERGAS index showed values of 0.72, 1.84 and 1.83 whereas the spatial ERGAS index exhibited slightly higher values of 1.3, 1.73, and 1.86, for pansharpen B-G-R-NIR combinations 1-1-1-1; 1-1-5-5 and 1-1-1-10, respectively. Being a value of 0 of each ERGAS index the maximum quality, the lower the ERGAS value, the higher the spectral quality of fused images. ERGAS error lower than 3 are considered as good quality for fused product [56]. In this study, the ERGAS errors obtained were considerably low (lower than 2 in all combinations), which implied a high spectral and spatial quality in the images obtained [57]. This premise is important in this type of study, as an accurate isolation of olive crowns need a very high spatial resolution but without losing the spectral information, especially in complex areas where a mixture of olive tree, natural cover, and soil spectral data can be observed. The increase of the ERGAS indexes values in the pansharpened image, where the red and near-infrared bands were over-weighed, was predictable. Gonzalo and Lillo-Saavedra [44] conceived the pansharpen algorithm performed in this study to apply the exact weighted factor to every band to obtain same spatial and spectral quality of the image, it means, to obtain the "best fused image". In this study, and controlling that the quality of the fused images does not exceed the ERGAS index limit, the weight of the bands was based on the necessity of the study but not in the control of the pansharpen quality.

**Table 2.** Spectral and spatial indexes to control the quality of the pansharpened images.

| Pansharpen Weight (B-G-R-NIR) | Spectral ERGAS | Spatial ERGAS |
| --- | --- | --- |
| 1-1-1-1 | 0.72 | 1.13 |
| 1-1-5-5 | 1.84 | 1.73 |
| 1-1-1-10 | 1.83 | 1.86 |

### 3.2. Segmentation

For the fifteen pansharpened fields analysed, three pansharpen combinations × five olive orchard fields, a considerable number of input parameters were tested to obtain the criteria that provided the most satisfactory inputs scenarios (Table 3). Each olive orchard field presented different characteristics, which involved different segmentation parameters. As the aim of the segmentation is to obtain objects with a similar or smaller area than an olive tree canopy, the values of the scale parameters were low, ranging from 12 in the fields A and E to 25 in the fields B and C. The pansharpen images with the weight 1-1-1-1 always required a smaller scale parameter than the other two pansharpen images. In the segmentation process, the spectral information (colour) presented more weight than the morphology of the objects (shape) in most of the scenarios evaluated, although colour never exceeded the 70% of the weight. Only in the field D, the percentage of both types of information was divided equally (50%).

As mentioned in Section 2.3, the compactness and smoothness characteristics of the morphology of the objects were fixed to 0.8 and 0.2, respectively.

**Table 3.** Most satisfactory segmentation parameters obtained for each pansharpened field.

| Field | Pansharpen Weight (B-G-R-NIR) | Scale Parameter | Colour | Shape | Compactness | Smoothness |
|---|---|---|---|---|---|---|
| A | 1-1-1-1 | 12 | 0.6 | 0.4 | 0.8 | 0.2 |
| | 1-1-5-5 | 20 | 0.7 | 0.3 | 0.8 | 0.2 |
| | 1-1-1-10 | 20 | 0.7 | 0.3 | 0.8 | 0.2 |
| B | 1-1-1-1 | 15 | 0.7 | 0.3 | 0.8 | 0.2 |
| | 1-1-5-5 | 25 | 0.7 | 0.3 | 0.8 | 0.2 |
| | 1-1-1-10 | 25 | 0.5 | 0.5 | 0.8 | 0.2 |
| C | 1-1-1-1 | 15 | 0.6 | 0.4 | 0.8 | 0.2 |
| | 1-1-5-5 | 25 | 0.7 | 0.3 | 0.8 | 0.2 |
| | 1-1-1-10 | 17 | 0.6 | 0.4 | 0.8 | 0.2 |
| D | 1-1-1-1 | 14 | 0.6 | 0.4 | 0.8 | 0.2 |
| | 1-1-5-5 | 14 | 0.5 | 0.5 | 0.8 | 0.2 |
| | 1-1-1-10 | 14 | 0.5 | 0.5 | 0.8 | 0.2 |
| E | 1-1-1-1 | 12 | 0.6 | 0.4 | 0.8 | 0.2 |
| | 1-1-5-5 | 19 | 0.7 | 0.3 | 0.8 | 0.2 |
| | 1-1-1-10 | 22 | 0.7 | 0.3 | 0.8 | 0.2 |

### 3.3. Olive Orchard Fields Classification

The accuracy assessments, OA and K, of the different pixel- and object-based classifications carried out in the different weighted pansharpened images of every field are displayed in Table 4. Figure 3 shows an example of the least and the most accurate olive crowns classifications in an individual field. Table 4 reveals slight differences between classifiers, yielding most of the scenarios evaluated very high classification accuracies and showing that olive trees canopies could be discriminated very accurately in most of the test carried out. All the classifiers achieved high comparable classification results, although some classifiers stood out above others. In pixel-based classifications, MP exhibited the highest average of OA and K with a value of 94.2% and 0.89, respectively. In object-based classification, MP obtained the highest accuracies values in eight of the 15 images analysed and DT obtained the highest precision in the remaining seven classifications. Nevertheless, the average results were slightly higher with the DT classifier offering OA and K values of 96.6% and 0.94%, while ML obtained values of 95.3% and 0.91%. ML and DT classification algorithms exhibited high reliability in all classifications performed, while SAM and MD yielded more erratic results, showing always the lowest accuracies of each analysis. As an example, in pixel-based classification of field E, MD obtained the highest OA values for the pansharpen combinations 1-1-5-5 and 1-1-1-10 with 93.4% and 95.2%, respectively, whereas this classifier showed very low OA value for the pansharpen combination 1-1-1-1, with a value of 77.3%. Nevertheless, all the olive orchard fields could be classified very accurately with at least one classifier, showing OA values greater than 90%. The results observed in this study satisfied the commonly accepted requirements to consider an accurate classification when the OA value is at least an 85% [58] and the Kappa coefficient exceeds the 0.75 [59]. Such high accuracies are essential if the olive orchard map obtained is going to be used in precision agriculture to design site specific management. To emphasise one classifier from all of them, ML classifier could be selected considering that yielded one of the highest precisions in all classifications performed and that the computational and expertise requirements involved in this classification method is lower than the other most accurate classifier, DT, a data mining algorithms which demands deeper knowledge. Additionally, ML algorithm is implemented in most of the image processing software.

**Table 4.** Classification accuracies of olive orchards at the five fields analysed in the three pansharpened images using different classification algorithms.

| Field | Image [1] | Pixel-Based | | | | | | | | Object-Based | | | | | | | |
|---|---|---|---|---|---|---|---|---|---|---|---|---|---|---|---|---|---|
| | | MD [2] | | SAM | | ML | | DT | | MD | | SAM | | ML | | DT | |
| | | OA [3] | K | OA | K | OA | K | OA | K | OA | K | OA | K | OA | K | OA | K |
| A | 1-1-1-1 | 94.4 | 0.91 | 92.6 | 0.88 | 95.0 | 0.92 | 91.7 | 0.86 | 96.2 | 0.93 | 96.6 | 0.94 | 97.8 | 0.96 | 94.9 | 0.91 |
| | 1-1-5-5 | 91.7 | 0.86 | 90.8 | 0.84 | 94.4 | 0.91 | 98.7 | 0.97 | 96.4 | 0.94 | 94.5 | 0.91 | 97.9 | 0.96 | 98.9 | 0.98 |
| | 1-1-1-10 | 89.4 | 0.82 | 88.0 | 0.79 | 92.8 | 0.87 | 98.6 | 0.98 | 91.7 | 0.86 | 95.7 | 0.93 | 98.8 | 0.98 | 98.7 | 0.98 |
| B | 1-1-1-1 | 91.3 | 0.83 | 94.2 | 0.89 | 98.7 | 0.97 | 95.7 | 0.91 | 88.8 | 0.78 | 91.3 | 0.82 | 99.1 | 0.98 | 96.9 | 0.95 |
| | 1-1-5-5 | 95.2 | 0.91 | 95.4 | 0.91 | 98.7 | 0.98 | 92.1 | 0.89 | 94.6 | 0.89 | 97.3 | 0.94 | 99.3 | 0.99 | 99.3 | 0.98 |
| | 1-1-1-10 | 95.7 | 0.86 | 86.5 | 0.73 | 97.5 | 0.95 | 94.3 | 0.90 | 94.1 | 0.88 | 77.9 | 0.59 | 98.7 | 0.97 | 98.5 | 0.97 |
| C | 1-1-1-1 | 90.4 | 0.81 | 93.4 | 0.87 | 97.7 | 0.95 | 96.5 | 0.94 | 88.8 | 0.78 | 88.4 | 0.77 | 97.5 | 0.95 | 97.0 | 0.94 |
| | 1-1-5-5 | 95.8 | 0.92 | 93.1 | 0.86 | 97.7 | 0.95 | 92.4 | 0.89 | 95.6 | 0.91 | 96.5 | 0.93 | 97.9 | 0.96 | 98.7 | 0.98 |
| | 1-1-1-10 | 95.5 | 0.91 | 85.3 | 0.71 | 96.6 | 0.93 | 94.1 | 0.90 | 94.9 | 0.90 | 64.9 | 0.31 | 98.5 | 0.97 | 98.6 | 0.97 |
| D | 1-1-1-1 | 69.7 | 0.39 | 75.4 | 0.51 | 88.1 | 0.76 | 84.8 | 0.80 | 78.9 | 0.58 | 86.1 | 0.72 | 79.7 | 0.59 | 87.2 | 0.82 |
| | 1-1-5-5 | 87.5 | 0.75 | 87.9 | 0.76 | 94.2 | 0.88 | 88.5 | 0.84 | 87.1 | 0.74 | 85.6 | 0.71 | 88.4 | 0.77 | 98.3 | 0.96 |
| | 1-1-1-10 | 89.5 | 0.79 | 85.3 | 0.71 | 92.3 | 0.85 | 87.3 | 0.83 | 85.7 | 0.72 | 88.0 | 0.76 | 91.6 | 0.83 | 98.7 | 0.97 |
| E | 1-1-1-1 | 77.3 | 0.55 | 76.7 | 0.54 | 86.7 | 0.73 | 86.8 | 0.83 | 82.0 | 0.64 | 79.2 | 0.59 | 89.4 | 0.79 | 89.0 | 0.84 |
| | 1-1-5-5 | 93.4 | 0.87 | 91.5 | 0.83 | 93.0 | 0.86 | 90.7 | 0.89 | 97.1 | 0.94 | 96.8 | 0.94 | 97.5 | 0.95 | 97.9 | 0.96 |
| | 1-1-1-10 | 95.2 | 0.91 | 88.1 | 0.76 | 89.9 | 0.80 | 88.8 | 0.85 | 96.1 | 0.92 | 82.1 | 0.64 | 97.3 | 0.95 | 97.1 | 0.95 |

[1] Pansharpen weight (B-G-R-NIR); [2] Method of classification: MD, Minimum Distance; SAM, Spectral Angel Mapper; ML, Maximum Likelihood; DT, Decision Tree; [3] Accuracy values: OA, overall accuracy (%); K, Kappa coefficient.

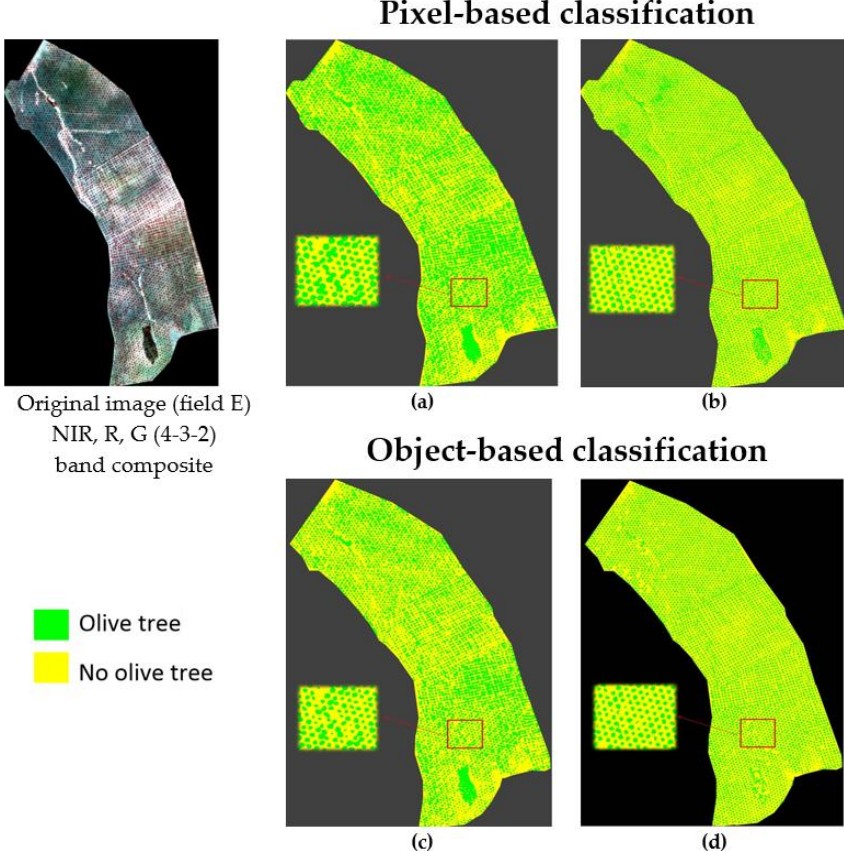

**Figure 3.** Result of the least (**a**,**c**) and most accurate (**b**,**d**) olive orchard classifications of field E.

Between pixel- and object-based analyses, no clear difference was observed. Despite that object-based classifications analysed a greater number of segmented variables than pixel-based, the precision of the classifications yielded were slightly higher. In object-based analyses, the increase of average OA values was approximately 1% for all classifiers except DT, which improved its general performance with an average increase of 4.5% and showed the most accurate average OA (96.6%). Twelve of the fifteen pansharpen combinations classified improved the accuracies when the object-based analysis were performed, but these increases usually were minimal, not exceeding in many cases the 1% of improvement. The most significant variations between object- and pixel-based analyses were observed in fields D and E. A 6.4% of improvement was observed in the pansharpen combination 1-1-1-10 of the field D, when the most accurate pixel-based analysis was performed with the MP classifier and yielded an OA value of 92.3%, whereas the most accurate object-based classification was obtained with DT algorithm and obtained an OA of 98.7%. Similarly, accuracy increases superior to 4% could be observed with the pansharpen combination 1-1-5-5 of both, field D and E, obtaining the greatest OA values of 98.3% and 97.9% with DT algorithm. Little advantage when applying object-based techniques can be observed in other studies focused on precision agriculture when pixel-based analyses offer reasonable performance. Pérez-Ortiz et al. [60] tested different scenarios of pixel- and object-based classifications to detect weeds in sunflower crops and obtained similar results, with improvements of up to 6%. Similarly, Castillejo González et al. [61] evaluated pixel- and object based classifications to distinguish late-season wild oat weed patches in wheat fields and suggested that the small sizes of the objects and the excellent behaviour of the classification algorithms in the pixel-based classifications did not produce a significant improvement over the precision obtained in the object-based classifications.

In object-based analyses eight different segmented variables were classified, which implies more information that can enhance or worsen the capacity to distinguish among categories. MD, SAM, and ML algorithms give equal weight to all the variables involved in the classification process, and use all these variables in the process, independently of the level of improvement or deterioration that the classification can suffer. Nevertheless, the DT algorithm analyses all the variables and selects only the information that really help to distinguish among categories, increasing its efficiency. Figure 4 shows the percentage of use for each variable in DT classifications. From the eight segmented variables managed in this study, DT algorithm only used six different variables in the total set of analyses, and only two or three were necessary in most of the DT classifications. All spectral variables were used in the DT analysis, but the bands that were selected more frequently were the NIR mean layer and the NDVI index. Whereas the NIR mean layer showed the highest level of intervention in DT analysis with a 44.4% for pansharpen combination 1-1-1-10 and a 37.5% for combination 1-1-5-5, the NDVI index band was used in the three pansharpen combinations with a 27.3%, 22.2%, and 12.5% of intervention for combinations 1-1-1-1, 1-1-1-10, and 1-1-5-5, respectively. From the five geometrical segmented variables, only Width, Length, and Border index were used in those classifications. With feature was the most useful, with a 45.5% of intervention for combination 1-1-1-1 and a 12.5% for combination 1-1-1-5. The scarce use of morphological variables can be explained because the olive is a tree that presents different canopy architecture depending on characteristics such as age, farm management, variety, pruning, etc. (Figure 5). The very high accuracies obtained with limited variables in DT classifications agrees with the idea exposed in [62], when they concluded that DT tends to be very efficient and robust when a large volume of predicted variables are introduced in a model, generally performing fast and being insensitive to noise in input data. This behaviour explain the rise of the accuracies that DT classifications exhibited in object-based analyses compared to ML algorithm.

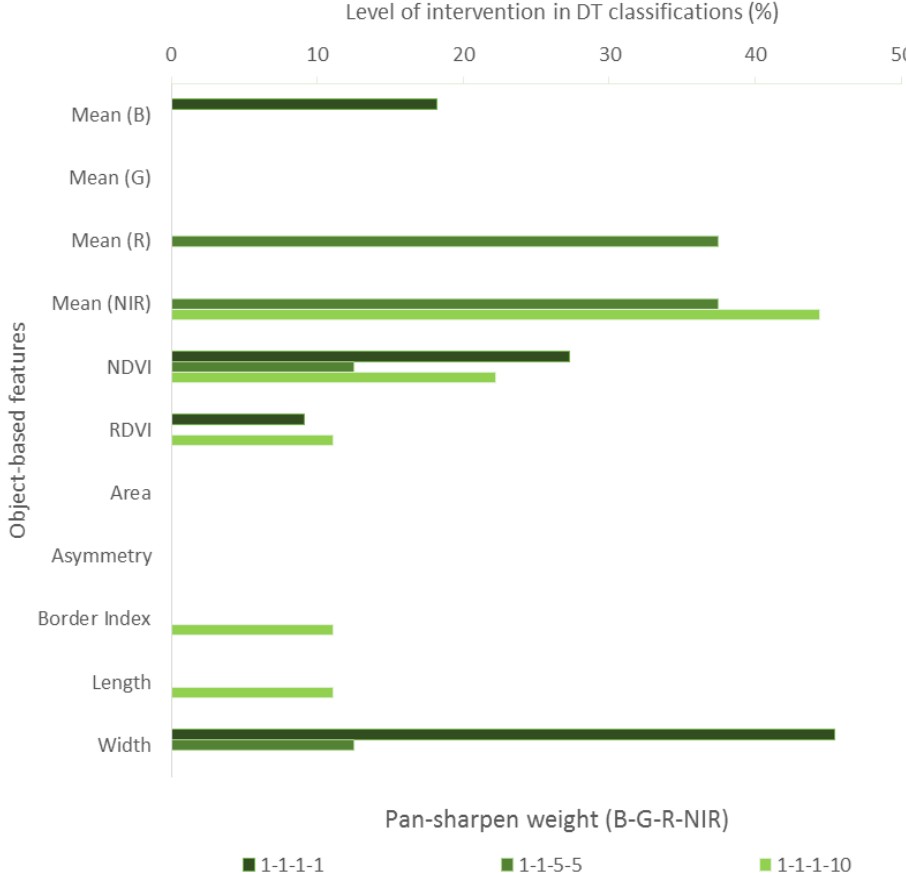

**Figure 4.** Relative contribution of object-based variables for DT classifications.

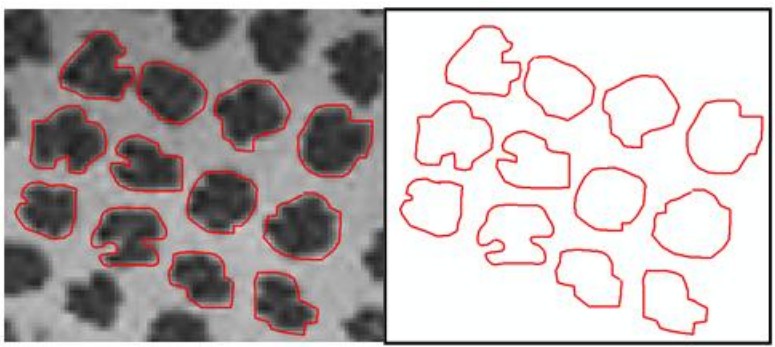

**Figure 5.** Example of differences in the morphology of the olive crowns.

The pansharpen process was necessary to obtain enough spatial resolution in the images to accurately distinguish the olive trees canopies. The original idea of alter the spectral information of the pansharpened images to emphasize the most useful spectral bands to distinguish vegetation from other land uses exhibited satisfactory results, offering increases of accuracy in three of the five fields studied. This improvement is especially significant in fields D and E, spectral complex fields which showed more difficult to isolate the olive trees. Field D showed the greatest improvement with the spectrally altered images. In pixel-based classifications, increases of approximately 6% and 4% were observed among the combination 1-1-1-1 (OA of 88.1%), and the combinations 1-1-5-5 (94.2%) and 1-1-1-10 (92.3%). More prominent were the increases observed in object-based classifications, where the combination 1-1-1-1 obtained an OA of 87.2% whereas the pansharpen combinations 1-1-5-5 and 1-1-1-10 showed OA values of 98.3% and 98.7%, respectively, which implied accuracy increases higher

than 11%. Similarly, field E showed important accuracy increases of 6% and 9% in pixel-based analyses and 8% and 7% in object-based classifications, for 1-1-5-5 and 1-1-1-10 pansharpened combinations, respectively. Although studies based on pansharpened images performed with different spectral weights were not found, the use of equal weight of the multispectral bands in the pansharpen process were evaluated in agronomic scenarios. When homogeneous land uses such as herbaceous crops were analysed with pansharpen imagery, the improvements in classification accuracy due to the use of more spectral and spatially detailed imagery could not really be considered as remarkable regarding the multispectral imagery [63]. Nevertheless, in land uses with high intraclass variability such as woody crops, where the individual trees must be isolated from soil and other covers presented in the field to design a site-specific crop management, the spatial detail is needed. García-Torres et al. [64] analyzed the capacity of different spatial resolutions to isolate olive trees and suggested that imagery with spatial resolutions from 0.25 to 1.5 m were generally suitable for olive grove characterization with olive trees of over 3–4 m$^2$. Scarce studies used pansharpen images to isolate trees, being more frequent to obtain the information of the tree based on the fusion of different types of information such as multispectral, hyperspectral or LiDAR data or with the analysis of very high spatial resolution UAV imagery. Johnson et al. [65] tested different pansharpened processes to map residential area trees and damaged oak trees. Since a hybrid approach including two pansharpening methods produced the best results in this study, they recommend users planning to process the pansharpen themselves rather than purchase pansharpened imagery directly from the image vendor, so that they can incorporate variations in the methodology for their analysis.

An accurate map of olive orchards fields enable the design of site-specific management treatments and can contribute to the follow up and assessment of agri-environmental regulations. Further studies will focus on establishing a hierarchical classification system that aims at discriminating all olive orchard fields present in the entire QB scene while evaluating the potential of image sharpening in classifying different land uses. In the first level of the hierarchical analysis, the olive orchard fields will be identified and isolated from the other land uses included in the whole studied region to, subsequently, characterize the olive trees of each field. Additionally, with each tree individually discriminated, different agri-environmental indicators of olive orchards related with the number and area of the trees (e.g., potential productivity of each tree and potential production of each plot) and related with bare soil and other vegetation covers (e.g., risk of erosion and run-off) can be predicted.

## 4. Conclusions

The olive production sector is characterized by a large number of small operators which directly affect production. The results of the present study show that pansharpened multi-spectral QuickBird imagery can be successfully used to map delineation of olive tree canopies. The knowledge of the accurate location and delimitation of the olive trees can be used as a basis for the precision management of fertilizers, pesticides and watering, since there is an obvious relationship between tree size and potential productivity based on the requirements of nutrients, watering doses and plant protection products such as fungicides. Although all classification algorithms tested offered accurate results, ML and DT were the two most precise classifiers. Knowing that the red and especially the near infrared bands are very important in the discrimination of vegetation, the alteration of the weight of these spectral bands in the pansharpen process enhanced significantly the accuracy of the olive trees delineation, with an average increase improvement of up to 9% and 11% in pixel- and object-based analyses, respectively.

With regard to recommending one methodology to define the olive trees crowns, two considerations should be made: the improvement in accuracy obtained and the computational or expertise requirements involved in the process. The decision of whether or not to carry out more complex analyses will depend on the importance of achieving maximum accuracy and the ratio of cost/efficiency wished to obtain in the objectives. If one aimed to create a crop inventory, then the performance of pixel-based classification with ML classifier and standard pansharpened images

would be the best choice. However, if it desires to produce a map that is ready to be used for precision agriculture decision-making procedures, object-based analyses with the DT algorithm and the pansharpened imagery with the near-infrared band altered would be highly recommended.

**Funding:** This research received no external funding.

**Acknowledgments:** I extend thanks to Javier Carrasco for his assistance with data processing.

**Conflicts of Interest:** The authors declare no conflict of interest.

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
