# Peer review of "Mapping of Olive Trees Using Pansharpened QuickBird Images: An Evaluation of Pixel- and Object-Based Analyses"

_agronomy, doi:10.3390/agronomy8120288_

Reviewer 1 Report

The focus of paper is on remote sensing techniques to map the olive trees using high resolution satellite imagery. However, authors didn’t consider the sate-of-the-art in this area f research. There are many unclear aspects in their research such as the reference data. The case study seems a very simple case where there is no spectral confusion (only wo classes).

1-      Title of the paper needs revision. It is usually said “based on” pixel and object based approaches.

2-      Line 14: how do you get “high spectral resolution” using QuickBird imagery that provides only 4 spectral bands?

3-      Line 17: it is “overall accuracy” and not “over”!

4-      Lines 5-57: this is not correct. There are bulk of research in the literature using high resolution satellite imagery for identification of tree species. For example you can see the following papers that you have not studied them:

Identification and Quantification of Tree Species in Open Mixed Forests using High Resolution QuickBird Satellite Imagery

Tree Species Classification with Random Forest Using Very High Spatial Resolution 8-Band WorldView-2 Satellite Data

Identification of tree species from high-resolution satellite imagery by using crown parameters

Feature selection for tree species identification in very high resolution satellite images

5-      Lines 65-66: how do you say this? You are going ahead of yourself!

6-      Lines 74-76: you should cite references.

7-      Table 1: why you haven’t used the spectral features such as NDVI for the pixel-based classification?

8-      What are your reference data for accuracy assessment?

9-      Your case study is so simple. You have just olive tree and bare soil? There is no spectral confusion here. Maybe you can get the same results with pan sharpening as well!

Author Response

I thank the reviewers for their valuable comments, which have enabled me to significantly improve this paper. I carefully revised the manuscript. Please find our responses to each individual comment below (attachment file).

Reviewer 2 Report

The manuscript reports a study on mapping of olive trees using pansharpened QuickBird images in pixel- and object-based analysis approaches. While the study deals with an interesting approach for classification of olive trees, I am wondering if the work could be of interest to the potential readers of Agronomy as well as fit well to the general scope of the journal. I have found several issues to be addressed before it is considered for publication in the journal. I recommend revising the title to represent what is actually examined in the study something like “mapping of olive trees using pansharpened QuickBird images in pixel- and object-based analysis approaches.” The authors compared the classification outcomes between the pansharpened QuickBird images in pixel- and object-based analysis approaches. However, it should be highlighted by comparing between multispectral QuickBird images and those pansharpened. Then, it might add a value to the current study. The overall compositions should be improved. Especially, methods applied should be described further in detail in the Materials and Methods section (i.e., 2.2 Data fusion). Citations should be made for NDVI and RDVI in Table 1. In addition, classification methods should be described using references in the subsection 2.4 Classification and accuracy assessments. I also think that wordings and phrases should be carefully reviewed for better articulate expressions. Some minor issues and comments are as follows.

Abstract: Recommend including a conclusion sentence.

Lines 47-48: monitor -> monitoring (?). I think “delivering or addressing” appears better. Consider to revise ‘obtain, detect, and determine’ with “‘obtaining, detecting, and determining”.

Line 51: The authors say that ‘herbaceous crops are easier to study with digital imagery”. What basis do the authors mention this? I do not agree with this speculation. It would be described with any proven previous works or a generally acceptable concept.

Line 64: ‘allow’ -> “allows”

Lines 67-76: References should be included in these descriptions.

Line 98: ‘is’ -> “was”

Line 103: ‘The study was…’ -> “The study sites were…”

Lines 104: ‘such us’ -> “such as”

Line 105: ‘vegetal cover’ -> “vegetation cover”

Lines 109-110: the figure caption: ‘study area’ -> ‘the study area”, ‘image’ -> “images”

Lines 112: ‘consist on’ -> “consist with”

Line 116: the distributor’s information needs to be included.

Line 154: ‘de size’ -> “the size” (?)

Line 157: ‘divided in’ -> “divided into”

Line 161: the pixels than…’ -> “the pixels that…”

Line 193: do you mean “…it is frequent”?

Line 198: ‘IJFUSION software’ -> “the IJFUSION software”

Line 199: ‘Madrid’ -> “Madrid, Spain”

Lines 201-202: Please use the current company name and the location of the ENVI software.

Line 204: ‘Results’ -> “Results and Discussion”

Author Response

I thank the reviewers for their valuable comments, which have enabled me to significantly improve this paper. I carefully revised the manuscript. Please find our responses to each individual comment below. (attachment file).

Round  2

Reviewer 1 Report

The authors addressed most of the comments. The title of the paper still needs some re-wordings (there are two "and" which is kind of repetitive). The figures labels also need some work to make the font size/type uniform throughout the paper.

Author Response

Once again I thank the reviewers for their valuable comments, which have enabled me to significantly improve this paper during the revision process. Please find the responses to each individual comment below.

Reviewer 1

Comments and Suggestions for Authors

The authors addressed most of the comments.

1-   The title of the paper still needs some re-wordings (there are two "and" which is kind of repetitive).

Thank you for the suggestion. The comment has been taken into account and the title was rewritten again. The new title is as follow

Mapping of olive trees using pansharpened QuickBird images: An evaluation of pixel- and object-based analyses

2-   The figures labels also need some work to make the font size/type uniform throughout the paper.

Really, the size of the figure labels was not the same for all figures of the manuscript. I have corrected all figure captions using the Palatino Linotype font and the size 9, as the template suggests. 

Reviewer 2 Report

Thanks for the efforts to improve the manuscript.

Author Response

Once again I thank the reviewers for their valuable comments, which have enabled me to significantly improve this paper during the revision process.

Reviewer 2

Comments and Suggestions for Authors

Thanks for the efforts to improve the manuscript.

Thank you for the suggestion and comments. They were very valuables.  
